# In the Subtropical Monsoon Climate High-Density City, What Features of the Neighborhood Environment Matter Most for Public Health?

**DOI:** 10.3390/ijerph17249566

**Published:** 2020-12-21

**Authors:** Wei Gao, Ruoxiang Tu, Hao Li, Yongli Fang, Qingmin Que

**Affiliations:** College of Forestry and Landscape Architecture, South China Agricultural University, Guangzhou 510640, China; gaowei@scau.edu.cn (W.G.); scautrx@scauladri.com (R.T.); lee19930808@gmail.com (H.L.); photon033@gmail.com (Y.F.)

**Keywords:** landscape architecture, environmental factor, correlation study, physical health, mental health, urban regeneration

## Abstract

Urbanization and climate change have been rapidly occurring globally. Evidence-based healthy city development is required to improve living quality and mitigate the adverse impact of the outdoor neighborhood environment on public health. Taking Guangzhou as an example to explore the association of neighborhood environment and public health and preferably to offer some implications for better future city development, we measured ten environmental factors (temperature (T), wind-chill index (WCI), thermal stress index (HSI), relative humidity (RH), average wind speed (AWS), negative oxygen ions (NOI), PM2.5, luminous flux (LF), and illuminance (I)) in four seasons in four typical neighborhoods, and the SF-36 health scale was employed to assess the physical and mental health of neighborhood residents in nine subscales (health transition(HT), physiological functions (PF), general health status (GH), physical pain (BP), physiological functions (RP), energy vitality (VT), mental health (MH), social function (SF), and emotional functions (RE)). The linear mixed model was used in an analysis of variance. We ranked the different environmental factors in relation to aspects of health and weighted them accordingly. Generally, the thermal environment had the greatest impact on both physical and mental health and the atmospheric environment and wind environment had the least impact on physical health and mental health, respectively. In addition, the physical health of the resident was more greatly affected by the environment than mental health. According to the results, we make a number of strategic suggestions for the renewal of the outdoor neighborhood environment in subtropical monsoon climate high-density cities and provide a theoretical basis for improving public health through landscape architecture at the neighborhood scale.

## 1. Introduction

In the process of global urbanization, with increasingly high population densities, public health has gradually become the most pressing urban problem. In terms of physical health, noncommunicable chronic diseases have become the leading cause of death for urban residents worldwide [1]. In terms of mental health, taking China as an example, the status of most urban residents (73.6%) is classified as sub-healthy [2]. In addition to age, income, education, gender, disease, nutrition, etc., the environment is a contextual determinant of health accounting for some disease burdens [3,4]. A large number of studies have shown that public health is directly related to the built environment [5,6].

Examining the relationship between the urban environment and environmental factors such as noise, light, temperature, humidity and air quality has yielded rich data in relation to the urban microclimate [7,8,9,10,11,12,13], mostly involving microclimate and green spaces. The thermal environment is closely associated with the increase in mortality, especially respiratory mortality [14,15,16,17]. The physiological effects of the high temperature include increasing body temperature, which affects water and salt metabolism, heat stroke, etc. [18]. Humidity is significantly related to the spread and mortality of infectious diseases such as influenza and hand-foot-mouth disease [19,20,21,22,23], which also exacerbates the negative impact of hot weather on mental health [24]. Air pollutants enter the human body mainly through respiration, so the respiratory system is the target organ for the airborne particles and other harmful substances [25]. The atmospheric environment also consists of some environmental factors that are beneficial to public health, such as negative oxygen ions and volatile substances from some plants [26]. The urban wind environment is closely related to the atmospheric environment, especially the air pollutants, which indirectly affects the public health of urban residents [27]. Ultraviolet radiation exposure to excessive light causes skin system diseases [28,29], while light pollution affects sleep and mental health problems [30]. Noise pollution affects the cardiovascular system [31,32], hearing impairment, endocrine system and mental health [33], etc., and even affects children’s cognitive ability and memory ability [32]. However, positive soundscapes help residents recover from stress [34] and improve residents’ well-being and self-assessment of health [35]. Most of the abovementioned studies have established the correlation between a single environmental factor and the health of residents, and few studies have evaluated the comprehensive health benefits of multiple variables in the community environment. 

The neighborhood environment is closely related to the residents’ health, and previous studies have found evidence that the neighborhood environment with high temperature and humidity has a negative impact on both physical and mental health of residents. The health-supportive environments could contribute to reducing negative environmental exposures and helping residents achieving a good quality of life [3,36]. In previous studies, access to green space has been proven to improve both physical and mental health [37,38], and the increase of land-use mix is associated with the reduction of residents’ obesity [39]. Another study found that exposure to neighborhood blue space is significantly associated with environmental harm reduction, stress reduction, social contact facilitation and elderly residents’ mental health [40]. However, there has been no prior study on the quantitative relationship between neighborhood environment and public health in subtropical monsoon climate areas.

As a first-tier city and a famous historical and cultural center located in the southern subtropics, Guangzhou experiences perennial high temperatures and humidity [41], with a large number of old neighborhoods in its historic urban areas and high population density [42]. Therefore, we selected Guangzhou to explore the relationship between public health and the neighborhood environment in high-density cities in subtropical monsoon climate areas. Research into the relationship between the neighborhood environment and public health in old neighborhoods allows us to examine links between landscape architecture and public health in high-density cities in order to provide more targeted, cost-effective, and viable foundations and guidance for effective urban renewal and transformation. Further, exploring the relative impacts of different environmental factors on public health is helpful to determine the key points and sequence of the urban neighborhood environment transforming, so as to provide residents with a more health-supportive living environment. Such research is also helpful for developing guidance on renewal of the outdoor environment and structures in high-density cities and subsequent evaluation of its success.

Taking Guangzhou as an example, this study demonstrates the relationship between neighborhood public health and the outdoor neighborhood environment in subtropical monsoon climate, high-density city. According to the relevant architectural design standards in China for studies of urban environmental physics, the urban physical environment is mainly divided into the thermal, wet, wind, atmospheric, light, and acoustic environments [43]. We consider each factor in the relationship between health and environment with respect to its relative importance (weight), form and mechanism. We then attempt to make strategic suggestions for the renewal of the outdoor neighborhood environment in high-density, old, urban neighborhoods in subtropical monsoon climate areas, providing a theoretical basis for landscape architecture to be used to improve and enhance public health at the neighborhood scale.

## 2. Materials and Methods 

### 2.1. Study Site

Guangzhou (23°6′ N, 113°15′ E) has a subtropical monsoon climate and was selected for this research because we consider it to be a typical city of subtropical monsoon climate areas (Figure 1). The annual average temperature in Guangzhou is 22.8 °C, the hottest month of the year is July, the coldest month is January, and the annual temperature range is 15.3 °C. On average, the daily maximum temperature exceeds 30 °C on 150.2 d of the year. Based on data from the past 30 years, the average annual relative humidity in the urban area is 73%, the annual rainfall is about 1906.8 mm, the longest continuous period of precipitation is generally in June, and the annual average wind speed is 1.5 m/s, making this a typical hot and humid area [41]. 

The four sample areas selected for the study were in the Liwan district and Yuexiu district, which are typical old neighborhoods in the historic city of Guangzhou (Figure 2). They have high population densities, an old built environment, undeveloped infrastructure, and a lack of public space.

### 2.2. Data Collection

#### 2.2.1. Environmental Factor Measurement

The major environment factors in the 4 sample areas were measured. Temperature (T), wind-chill index (WCI), and thermal stress index (HSI) were selected to represent the thermal environment of the sample areas; relative humidity (RH) to represent the wet environment; average wind speed (AWS) to represent the wind environment; negative oxygen ions (NOI) and PM2.5 to represent the atmospheric environment; luminous flux (LF) and illuminance (I) to represent the light environment; and noise (N) to represent the sound environment. These environmental factors comprehensively reflect the environment in the study plots. In this study, T, RH, AWS, WCI, and HSI were measured with a handheld weather station (KESTREL 3000); I was measured using a portable colorimeter (TES-136); LF was measured with a portable luminous flux meter (TES-133); N was measured with a noise meter (ES-1357); NOI was measured with an air positive and negative oxygen ion detector (KEC-900 II); and PM2.5 was measured with a Laser Dust Measuring Instrument (KornoGT-1000). 

Guangzhou is hot and humid most of the time [41]. The city experiences a very short spring and autumn, and the climate primarily alternates between summer and winter conditions. Measurements were taken in December 2018, March 2019, July 2019, November 2019, and January 2020. Two consecutive sunny days were selected in each quarter, and the daily measurement period was from 8:00 a.m. to 8:00 a.m. once every 2 h. Five sample points distributed around the center of each neighborhood were selected. The results are summarized in Table 1.

#### 2.2.2. Questionnaire Distribution and Recovery

According to past research, the SF-36 scale has good reliability and validity for Chinese and Guangzhou residents and can accurately reflect their health [44,45]. The SF-36 health scale contains 9 subscales, 8 of which measure physical health and mental health, and there is a separate health change self-assessment (Health Transition, HT) subscale to measure overall change in health status [46]. 

The remaining 8 subscales assess 8 health concepts: Limitations in physical activities because of health problems; limitations in social activities because of physical or emotional problems; limitations in usual role activities because of physical health problems; bodily pain, psychological distress, and well-being; limitations in usual role activities because of emotional problems; energy and fatigue; and general health perceptions [47]. Physical health includes 4 subscales: Physiological functions (PF), general health status (GH), physical pain (BP), and physiological functions (RP). Mental health includes 4 subscales: Energy vitality (VT), mental health (MH), social function (SF), and emotional functions (RE). In this study, random sampling was used to gain a picture of the physical and mental health status of residents in the 4 neighborhoods using the SF-36 scale.

Within 2 weeks after each time, the environmental factors were measured in each area by distributing 60 questionnaires to each neighborhood, i.e., a total of 1160 questionnaires distributed on 5 occasions. All questionnaires were recovered. Four incomplete responses were excluded, and finally, 1156 were used in the statistical analysis. There were 576 males (49.83%) and 580 females (50.17%). The age range was 12 to 89 years, with an average of 48.51 years (Table 2). The basic score for each subscale of the SF-36 scale is shown in Table 3.

### 2.3. Statistical Analysis

Evaluation of reliability and validity of the SF-36 scale [48]:

(1) Split-half reliability: Items were divided into 2 groups, each containing the same number, according to whether they had odd or even serial numbers. The scores for each group were calculated, and their Pearson correlation coefficient *R*_h_, was used to calculate the split-half reliability *R* of the whole scale: (1)$$R=\frac{2R_{h}}{1+R_{h}}$$

(2) Internal consistency reliability: We calculated the Pearson correlation coefficient and Cronbach’s α coefficient between subscales. Cronbach’s α was calculated using the formula:(2)$$\alpha =\frac{k}{k-1}(1-\frac{\sum S_{i}{}^{2}}{S_{T}^{2}})$$
where *k* is the number of entries contained in each subscale, *S_i_*^2^ is the variance of each item in the subscale, and *S_T_*^2^ is the total variance of the subscale.

(3) Content validity: The correlation coefficients between each subscale and the correlation coefficient between each subscale and the total score were used to evaluate the content validity.

(4) Construct validity: The common factors of each characteristic with root >1 in the 8 subscales of the SF-36 scale were extracted by factor analysis and the common variance of each subscale was calculated. Then, the value of the load of each subscale on the common factor was calculated by the maximum variance rotation method. The construct validity for each was evaluated by comparison with the theoretical model.

As there are many factors affecting human health, in order to explore the experimental data, we used the linear mixed effect model in an analysis of variance:*y* = *X**β* + *Z**u* + *ε*(3)
where *y* is the total score in the SF-36 scale, *β* represents fixed effects (neighborhood, season, neighborhood × season), *u* represents random effects (sex, age), *ε* is the error, *X* is the design matrix of fixed effects, and *Z* is the design matrix of random effects.

The lme4 package [49] in R version 4.0 [50] was used to fit the linear mixed effect model, and the sjstats and lmerTest packages [51] were used to test the significance of the linear mixed effect model. The Pearson correlation analysis, significance testing, and factor analysis were conducted using the psych package [52], and path analysis was conducted using the agricolae package [53] in R version 4.0. 

## 3. Results

### 3.1. Reliability and Validity Analysis of the SF-36 

#### 3.1.1. Reliability Analysis of the SF-36 

(1) Split-half reliability: The Pearson correlation coefficient (*R*_h_) of the sum of the two data subgroups was 0.86 (*p* < 0.001), and the split-half reliability of the scale was *R* = 0.9274.

(2) Internal consistency reliability: The overall Cronbach’s α coefficient of the SF-36 scale was 0.82, and the Cronbach’s α coefficients of BP, PF, RP, GH, VT, SF, RE, and MH were 0.80, 0.90, 0.80, 0.82, 0.75, 0.81, 0.73, and 0.74, respectively (all *p* < 0.01).

The split-half reliability of the SF-36 scale was >0.9, and the Cronbach’s α coefficient for each subscale and overall were more than 0.7, which indicates that the scale has high reliability, good overall internal consistency, and a stable detection function.

#### 3.1.2. Validity Analysis of the SF-36

(1) Content validity: The correlation between each subscale and the total score was highly significant (*p* < 0.001). With the exception of the correlation between mental health and health changes, the correlations between other subscales were also highly significant (*p* < 0.001) (Table 4). Thus, the scale has credible content validity.

(2) Construct validity: Factor analysis was carried out for each subscale, and two common factors with values >1 were extracted for analysis. Together, these two common factors explained 62.32% of the total variance. The load values of each subscale with respect to the two common factors, according to the maximum variance rotation method, showed that the five subscales BP, PF, RP, GH, and SF were related to the first factor, F1. VT, MH, and RE were related to the second factor, F2 (Table 5). BP, PF, GH, VT, and MH of the actual model were completely consistent with the theoretical model, and RP and SF were fairly consistent. Only RE was not consistent with the theoretical model. This confirms that the results were basically in line with the design concept of the scale.

### 3.2. Analysis of Variance

Using the linear mixed model, taking neighborhood, season, and their interaction as fixed effects and gender and age as random effects, the total scores of health subscales in SF-36 were examined by analysis of variance. The results are presented in Table 6 and Table 7. The analysis showed that the effects of season on the fixed effects and the interaction between neighborhood and season on the health status of neighborhood residents were highly significant (*p* < 0.001). For the random effects, the effects of age and gender on the health status of neighborhood residents were highly significant (*p* < 0.001). The difference between season and the neighborhood × season interaction with respect to the fixed effects indicates that different characteristics of the environment have different effects. Thus, in order to explore how the environment affects health, it is necessary to examine the correlation between environmental factors and health subscales.

### 3.3. Correlation between Environmental Factors and Health

#### 3.3.1. Correlation between Health Subscales and Environmental Factors

From the results of the analysis of variance, we can see that the health of residents was significantly affected by the environment. In order to explore how environmental factors affected health, correlations between the health subscales and environmental factors were calculated (Figure 3).

GH, BP, and VT exhibited highly significant correlations with environmental factors; MH, RE, HT, SF, and PF were significantly correlated with environmental factors; and RP was not correlated with environmental factors. The ranking of the correlations between health subscales and the environment from high to low was GH > BP > VT > MH > RE > HT > SF > PF > RP.

In terms of environmental factors, T, WCI, HSI, RH, and NOI had highly significant effects on health and LF, I, WAS, PM2.5, and N had significant effects. The ranking of correlations between environmental factors and health from high to low was T > WCI > HIS > LF > RH > NOI > I > WAS > PM2.5 > N.

#### 3.3.2. Correlation between Health Types and Environmental Factors

The physical health score (HI1) and mental health score (HI2) of residents were calculated on the basis of the factors shown in Table 5, and the correlation analysis was conducted using the total score (SFTOTAL), HI1, HI2, and the environmental factors for each group. The results are shown in Figure 4.

SFTOTAL and physical health HI1 were very significantly affected by HSI, T, and WCI and significantly affected by N and AWS. HI2 was very significantly affected by T and WCI and significantly affected by HSI. HI1 was more greatly affected by the environment than HI2.

From the perspective of environmental factors, the effect of T, WCI, and HSI on HI1 and HI2 was highly significant. The effects of N and AWS on HI1 were significant, while the effects of other environmental factors on health were not significant. The ranking of the correlation between environmental factors and HI1 from high to low was T > WCI > HIS > N > AWS > RH > I > LF > PM2.5 > NOI. The ranking of the correlation between environmental factors and HI2 from high to low was T > WCI > HIS > N > I > LF > RH > NOI > AWS > PM2.5. Finally, the ranking of the correlation between environmental category and health from highest to lowest was thermal environment > acoustic environment > wind environment > wet environment > light environment > atmospheric environment. Similarly, the ranking for mental health was thermal environment > acoustic environment > light environment > wet environment > atmospheric environment > wind environment.

#### 3.3.3. Path Analysis of Environmental Factors in Relation to Health 

Based on the correlation between environmental factors and health types, the path coefficient can be calculated by the Gauss-Doolittle method. The results are shown in Table 8.

The analysis revealed that I had the greatest effect on SFTOTAL, followed by LF. The most important influence on HI1 was HSI, followed by WCI. The most important influence on HI2 was I, followed by T. For R2, the environmental factor that made the greatest contribution to SFTOTAL was WCI, followed by I. The environmental factor that made the greatest contribution to HI1 was HSI, followed by WCI, and the greatest contribution to HI2 was T, followed by I.

## 4. Discussion

### 4.1. There Is a Close Correlation between Public Health and the Outdoor Neighborhood Environment in Subtropical Monsoon Climate and High-Density Cities

Our analysis revealed a significant, sometimes extremely significant, correlation between environmental factors and the health status of neighborhood residents with the change of seasons in the old neighborhood of Guangzhou. The key aspects of health affected included GH (reflecting residents’ health status and its development trend), BP (reflecting the degree of physical pain and the impact of pain on daily life), and VT (reflecting the subjective feelings of residents about their own energy and fatigue). The correlation between these health subscales and environmental factors was highly significant. In the same region, for example, in Guangzhou and Hong Kong, some studies have also proven that the neighborhood environments are closely related to mental health [40,54,55] and that perceived neighborhood environments are positively associated with sense of community and self-rated health [56]. These results suggest that, in subtropical monsoon climate areas, there is a close correlation between public health and the outdoor neighborhood environment in old, high-density, urban neighborhoods. The result support the impact of the neighborhood environment on the residents’ health [57].

The data also suggest that the neighborhood environmental renewal and urban regeneration of such neighborhoods could play a very important role in improving public health. The current researches support that neighborhood renewal promote physical activity and mental health, reduce obesity and smoking [58,59,60], and promote health equity between regions [61], which is often small and positive [62]. There is currently little robust evidence showing that neighborhood renewal provides health gains beyond mainstream service provision [61,62]. For example, neighborhood renewal cannot improve respiratory problems caused by the wet environment and wind environment [60]. However, the results may be affected by regional characteristics and climate. Our research results provide theoretical support for health benefit of community renewal. Through evidence-based design, neighborhood renewal can improve the environmental factors of the community by changing the built environmental factors, and further provide support improvement for the residents’ health. Renovating the urban built environment with public health improvement as the goal can deliver good results in high-density cities. 

### 4.2. Selecting Priorities for Environmental Renewal Work in Old Neighborhoods with the Intention of Improving Public Health Can Be Informed by Weighting the Different Environmental Factors

Based on our data, thermal and acoustic environments appear to have the greatest impact on physical health and mental health and should be prioritized in urban renewal work in subtropical monsoon climate areas. Through the control of indexes such as the greening rate, green coverage rate, and the sky visual factor, and through appropriate design of solar radiation areas [63] and the placement of water features, the urban heat island effect [64] can be alleviated, the temperature of urban public spaces can be adjusted, and thermal comfort can be improved. Traffic planning and dynamic zoning, combined with strategies such as adding street trees, hedges, and other design elements for sound insulation, can deliver a low-noise environment and a pleasant sound experience [65], and this should be a priority in the renewal of old neighborhoods. The wind, wet, and light environments are secondary with respect to such renewal when attempting to deliver public health outcomes. The atmospheric environment is the least important because the scale of old neighborhoods tends to be relatively small, and neither the perception of residents nor the specific design strategy can have a significant correlation with the broader atmospheric environment. In previous studies, researchers have found evidence that the neighborhood environment with high temperature and humidity has a negative impact on both the physical and mental health of residents [14,15,19,24]. There are some epidemiological evidences that people living in hotter urban areas are at higher risk of morbidity or mortality from higher ambient temperatures [10], while children are more susceptible to environmental factors, especially heat and air pollution [66]. Another study in tropical coastal cities found that wind speed received the highest risk severity value among other microclimate factors [67]. Of all the environmental types, the wind environment needs to be discussed separately because, except for the prevailing wind in the city, the speed of the near-ground wind in old neighborhoods is usually gentle, and the “wind shadow area” is basically in a windless state [68]. The internal wind environment in the old neighborhood is mainly the result of thermal circulation, and the measured wind speeds are all low [69]. The average wind speed measured in this study only represents the wind conditions in the study areas and not the complete urban wind environment. Therefore, the relative importance of the wind environment in this study does not represent the actual influence of wind environment on health. On the other hand, the urban wind environment directly affects the thermal, humid, and atmospheric environments by affecting somatosensory temperature [70,71], relative humidity, and air pollutants such as PM2.5 [72]. In other words, the relative effects of the thermal, humid, and atmospheric environments actually result from their interactions with the urban wind environment. In this study, the indirect effect of average wind speed on SFTOTAL accounted for 44.2% of the correlation coefficient (Table 8). Therefore, design strategies such as maintaining ventilation corridors, creating microtopography to direct the breeze, and controlling the density and height of trees and shrubs should be given the highest priority in order to improve the wind, thermal, wet, and atmospheric environments, and, at the same time, maximize landscape performance. The above analysis indicates that the neighborhood-scale wind environment needs to be considered from two perspectives, the neighborhood wind environment and urban wind environment, confirming the need for urban microclimate research with the wind environment as the main focus.

To sum up, in subtropical monsoon climate areas, the priorities during renewal of old, high-density, urban neighborhoods can be divided into three levels of priority: First, the urban wind, thermal, and sound environments; second, the neighborhood wind, wet, and light environments; and third, the atmospheric environment.

### 4.3. During Environmental Renewal of Old Neighborhoods, Weighting the Relative Importance of Different Factors in Relation to Health Can Be Helpful When Identifying Projects to Prioritize

Our data show that physical health is more likely to be affected by the environment than mental health in the old neighborhoods of Guangzhou. Thus, providing recreation, and sports facilities [73], with the goal of reducing pain and improving physiological function, are key in any environmental renewal work.

We found that physical health was more influenced by the wind environment than mental health, which showed a greater response to the light environment. The wind environment is directly related to the spread of airborne infectious diseases, such as influenza [19], and is also related to the development of noncommunicable chronic diseases, such as rheumatism. Thus, evidence-based design in relation to the wind environment from the perspective of disease pathology and public safety is important in order to deliver physiological and health support via the built environment [67]. By designing a comfortable light environment, for example, with appropriate selection of materials in areas that receive direct sunlight, using trees with low crowns to provide shade, and choosing diffuse lights to avoid glare, renewal projects can support improved mental health.

BP and VT are two health subscales which are very significantly affected by environmental factors, suggesting that special attention should be paid to renewal projects that could deliver physical pain relief (a high priority for physical health) and restoration of vitality (a high priority for mental health). In summary, during urban renewal in old neighborhoods, projects that deliver physical and mental health recovery support should be prioritized, followed by those that improve health in general. Providing pain relief and recovery services is a particularly important goal for the urban renewal of old neighborhoods.

## 5. Conclusions

In conclusion, this empirical research proved that the neighborhood environment is related to the residents’ public health in subtropical monsoon climate and high-density cities. We measured 10 environmental factors in four seasons in four typical communities, and the SF-36 health scale was employed to assess the physical and mental health of community residents in 9 subscales. The linear mixed model was used in an analysis of variance and revealed a strong correlation between public health and the outdoor neighborhood environment. We explored the correlation between public health and the outdoor neighborhood environment of old communities in high-density cities in subtropical monsoon climate regions and demonstrated the relationship between landscape architecture and community public health. We found that the thermal environment had the greatest impact on both physical and mental health, and the physical health of residents was more greatly affected by the environment than mental health. Especially, we make a number of strategic suggestions for the renewal of the outdoor neighborhood environment according to those results above, which can provide a theoretical basis for improving public health through landscape architecture at the community scale. However, the data volume affected by residents’ willingness to cooperate can be still improved in the future study. Additionally, the variables in this paper have not yet involved the space form of the neighborhood environment, the distribution of facilities, and other built environment factors. The correlation between the neighborhood built environment and the neighborhood outdoor environment has not been revealed. 

In future research, we plan to further measure the built environment factors of the neighborhoods. With more research data for analysis, the impact of the neighborhood’s outdoor built environment on environmental factors and residents’ public health can be analyzed and more instructive neighborhood renewal strategies can be summarized.

## Figures and Tables

**Figure 1 ijerph-17-09566-f001:**
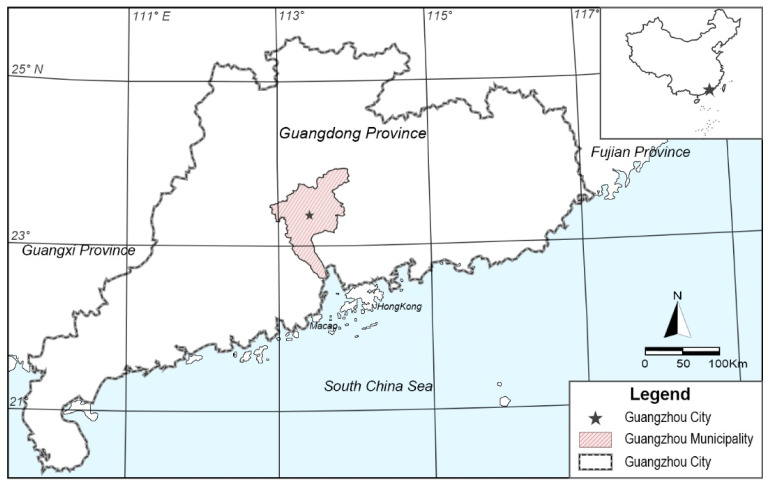
Location of Guangzhou.

**Figure 2 ijerph-17-09566-f002:**
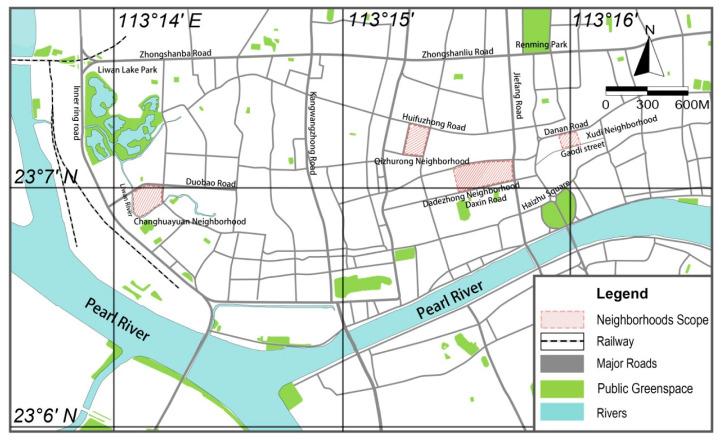
Distribution of the four survey areas in Guangzhou.

**Figure 3 ijerph-17-09566-f003:**
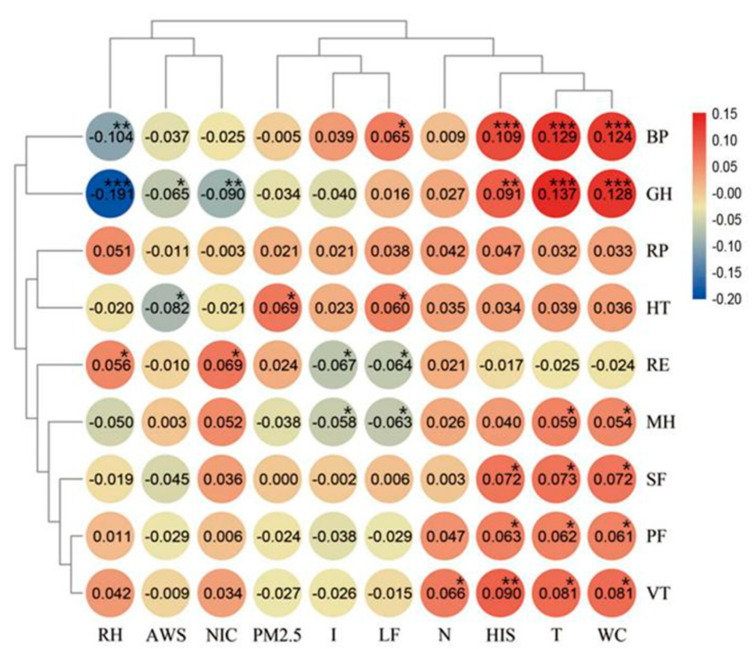
Correlation between environmental factors and health subscales. Note: “***” indicates a significant correlation at the 0.001 level, “**” indicates a significant correlation at the 0.01 level, “*” indicates a significant correlation at the 0.05 level.

**Figure 4 ijerph-17-09566-f004:**
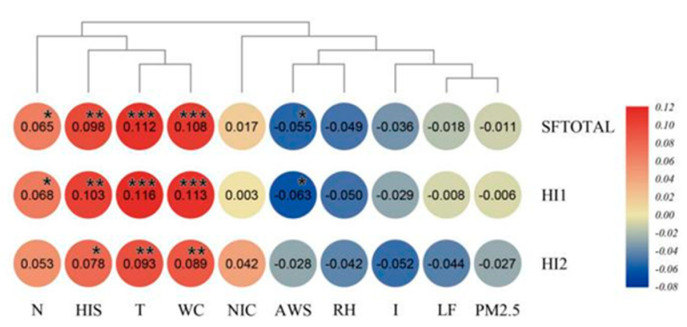
Correlation between total score, mental health, physical health, and environmental factors. Note: “***” indicates a significant correlation at the 0.001 level, “**” indicates a significant correlation at the 0.01 level, “*” indicates a significant correlation at the 0.05 level.

**Table 1 ijerph-17-09566-t001:** Descriptive statistics for the environmental factors in the sample plots.

Environmental Factors	Minimum	Maximum	Mean	Standard Deviation	Coefficient of Variation (%)
N (db)	50.46	61.55	54.96	3.63	6.61
I (Lm)	901.87	17,482.34	5215.39	3734.67	71.61
LF (Lx)	0.90	15.16	4.40	3.04	69.05
AWS (m/s)	0.12	0.91	0.53	0.22	40.92
T (°C)	16.22	33.80	25.07	5.14	20.51
WCI	16.24	33.83	25.08	5.34	21.28
RH (%)	60.55	83.21	69.95	7.02	10.04
HSI	16.15	44.89	28.03	9.03	32.23
NOI	−2.56	−0.44	−1.41	0.64	45.88
PM2.5 (μg/m^3^)	23.42	88.80	51.59	19.09	37.01

**Table 2 ijerph-17-09566-t002:** Gender and age distribution of the respondents.

Age Range (Year)	12–19	20–29	30–39	40–49	50–59	60–69	70–79	80–89
Male	31	95	78	90	135	95	37	17
Female	23	77	78	88	118	88	64	44

**Table 3 ijerph-17-09566-t003:** SF-36 Basic scores for each subscale of the scale.

Health Subscales	Minimum	Maximum	Mean	Standard Deviation	Coefficient of Variation (%)
BP	21.00	100.00	82.91	16.74	20.20
PF	0.00	100.00	84.71	19.42	22.92
RP	0.00	100.00	76.67	33.47	43.65
GH	5.00	100.00	65.47	20.65	31.54
VT	10.00	100.00	72.92	16.74	22.96
SF	0.00	112.50	93.03	20.97	22.54
RE	0.00	100.00	77.22	33.75	43.71
MH	4.00	100.00	72.83	16.73	22.97
HT	0.00	100.00	50.67	21.02	41.47

**Table 4 ijerph-17-09566-t004:** Correlation coefficients between subscales and between subscales and total scores.

Health Subscales	BP	PF	RP	GH	VT	SF	RE	MH	HT
PF	0.53 ***								
RP	0.46 ***	0.49 ***							
GH	0.53 ***	0.52 ***	0.44 ***						
VT	0.38 ***	0.43 ***	0.38 ***	0.46 ***					
SF	0.56 ***	0.52 ***	0.47 ***	0.51 ***	0.52 ***				
RE	0.30 ***	0.20 ***	0.47 ***	0.24 ***	0.30 ***	0.40 ***			
MH	0.26 ***	0.21 ***	0.24 ***	0.32 ***	0.59 ***	0.44 ***	0.34 ***		
HT	0.34 ***	0.29 ***	0.27 ***	0.40 ***	0.20 ***	0.25 ***	0.11 ***	0.07 *	
Total	0.68 ***	0.73 ***	0.62 ***	0.78 ***	0.77 ***	0.75 ***	0.46 ***	0.66 ***	0.39 ***

Note: “***” indicates a significant correlation at the 0.001 level, “*” indicates a significant correlation at the 0.05 level.

**Table 5 ijerph-17-09566-t005:** SF-36 practical and theoretical model factor load.

Subscales	Actual Model		Theoretical Model	
	F1	F2	Physiological Factors	Psychological Factors
BP	0.71	0.17	≥0.70	r ≤ 0.30
PF	0.72	0.13	≥0.70	r ≤ 0.30
RP	0.64	0.16	≥0.70	r ≤ 0.30
GH	0.66	0.24	0.30 < r < 0.70	0.30 < r < 0.70
VT	0.46	0.55	0.30 < r < 0.70	0.30 < r < 0.70
SF	0.67	0.36	0.30 < r < 0.70	≥0.70
RE	0.36	0.38	r ≤ 0.30	≥0.70
MH	0.12	0.99	r ≤ 0.30	≥0.70

**Table 6 ijerph-17-09566-t006:** ANOVA results for the fixed effects in the linear mixed model.

Fixed Effects	Sum Square	Mean Square	Degree of Freedom	Denominator Degree	F Value	*p* Value
Neighborhood	1177.00	392.34	3	911.83	1.96	0.117027 ^ns^
Season	3130.70	782.67	4	918.16	3.92	0.003613 **
Neighborhood × season	4079.30	509.91	8	909.80	2.55	0.009220 **

Note: “**” indicates a significant difference at the 0.01 level. “^ns^” indicates the difference is not significant.

**Table 7 ijerph-17-09566-t007:** ANOVA results for the random effects in the linear mixed model.

Random Effects	Number of Parameters	Logical Value	Red Pool Information Rules	Degree of Freedom	*p* Value
<none>	19	−3937.8	7913.5		
Age	18	−3965.6	7967.3	1	8.175 × 10^−14^ ***
Gender	18	−3941.9	7919.8	1	0.003951 **

Note: “***” indicates a significant difference at the 0.001 level, and “**” a significant difference at the 0.01 level.

**Table 8 ijerph-17-09566-t008:** Path analysis of health types by environmental factors.

Health Types	Environmental Factors	Correlation Coefficient	Direct Effects	R2 Total Contribution of Environmental Factors	Indirect Effects
Total	Adoption of N	Adoption of I	Adoption of LF	Adoption of AWS	Adoption of T	Adoption of WC	Adoption of RH	Adoption of HIS	Adoption of NIC	Adoption of PM2.5
SFTOTAL	N	0.0651	0.0466	0.003	0.0185		0.1313	−0.0941	0.0118	0.0031	0.0161	0.0044	−0.0106	−0.0075	−0.036
I	−0.0362	−0.4489	0.0163	0.4127	−0.012		0.3651	−0.0126	0.0168	0.0947	−0.0035	−0.0702	−0.0075	0.0419
LF	−0.0183	0.3781	−0.0069	−0.3964	−0.0104	−0.4391		−0.0102	0.0171	0.0968	−0.0044	−0.0702	−0.0103	0.0344
AWS	−0.0552	−0.0308	0.0017	−0.0244	−0.0191	−0.2218	0.1506		0.0035	0.0242	0.004	−0.0291	−0.0028	0.0662
T	0.1121	0.037	0.0042	0.0751	0.0037	−0.2173	0.1844	−0.0026		0.2016	−0.0073	−0.1284	0.0031	0.0377
WC	0.1081	0.1997	0.0216	−0.0916	0.0033	−0.2128	0.1807	−0.0031	0.0349		−0.0069	−0.1298	0.0034	0.0385
RH	−0.0493	0.0176	−0.0009	−0.0669	0.0108	0.0951	−0.0979	−0.0061	−0.015	−0.0827		0.0318	−0.0028	0
HIS	0.0982	−0.1342	−0.0132	0.2324	0.0033	−0.2399	0.1995	−0.0056	0.0339	0.1976	−0.004		0.0025	0.0452
NIC	0.0172	0.0285	0.0005	−0.0113	−0.01	0.1086	−0.1242	0.0023	0.0035	0.0222	−0.0015	−0.0106		−0.0017
PM2.5	−0.0113	−0.0851	0.001	0.0738	0.0179	0.2264	−0.1543	0.0202	−0.0157	−0.0927	0	0.0715	0.0006	
HI1	N	0.0683	0.0215	0.0015	0.0468		0.1517	−0.1519	0.0714	0.0553	0.0821	0.0886	−0.1261	−0.013	−0.1113
I	−0.0292	−0.5222	0.0152	0.493	−0.0067		0.5892	−0.0761	0.2951	0.4822	−0.0716	−0.8356	−0.013	0.1295
LF	−0.0081	0.6094	−0.0049	−0.6175	−0.0058	−0.5073		−0.0621	0.3013	0.4924	−0.0886	−0.8356	−0.0179	0.1062
AWS	−0.0631	−0.1583	0.01	0.0952	−0.0107	−0.2563	0.243		0.0615	0.1231	0.0818	−0.3469	−0.0049	0.2046
T	0.1161	0.6109	0.0709	−0.4948	0.0021	−0.251	0.2977	−0.0155		1.0259	−0.1465	−1.5293	0.0054	0.1165
WC	0.1134	1.0293	0.1167	−0.9159	0.0019	−0.2458	0.2916	−0.0186	0.6148		−0.1397	−1.5451	0.006	0.1191
RH	−0.0502	0.3406	−0.0171	−0.3908	0.006	0.1098	−0.1579	−0.0373	−0.2644	−0.4206		0.3784	−0.0049	0
HIS	0.1032	−1.5734	−0.1624	1.6766	0.0019	−0.2772	0.3219	−0.0341	0.5964	1.0054	−0.0818		0.0043	0.1398
NIC	0.0031	0.0538	0.0002	−0.0507	−0.0056	0.1255	−0.197	0.014	0.0615	0.1129	−0.0307	−0.1261		−0.0052
PM2.5	−0.0062	−0.2551	0.0016	0.2489	0.01	0.2615	−0.2491	0.1226	−0.2767	−0.4719	0	0.8514	0.0011	
HI2	N	0.0532	0.0769	0.0041	−0.0237		0.1153	−0.0798	0.0087	0.0214	−0.0011	0.0012	−0.0142	−0.0147	−0.0605
I	−0.0521	−0.3995	0.0208	0.3474	−0.0214		0.3096	−0.0093	0.1143	−0.0064	−0.001	−0.0942	−0.0147	0.0704
LF	−0.0441	0.315	−0.0139	−0.3591	−0.0184	−0.3855		−0.0076	0.1166	−0.0065	−0.0012	−0.0942	−0.0201	0.0577
AWS	−0.0282	−0.0171	0.0005	−0.0111	−0.0339	−0.1947	0.1277		0.0238	−0.0016	0.0011	−0.0391	−0.0055	0.1112
T	0.0933	0.2413	0.0225	−0.148	0.0066	−0.1908	0.1564	−0.0019		−0.0136	−0.0019	−0.1723	0.0061	0.0634
WC	0.0892	−0.0144	−0.0013	0.1036	0.0059	−0.1868	0.1532	−0.0023	0.238		−0.0019	−0.1741	0.0067	0.0648
RH	−0.0421	0.0024	−0.0001	−0.0445	0.0192	0.0835	−0.083	−0.0045	−0.1024	0.0056		0.0426	−0.0055	0
HIS	0.0782	−0.1794	−0.014	0.2576	0.0059	−0.2106	0.1691	−0.0042	0.2309	−0.0133	−0.0011		0.0049	0.076
NIC	0.0423	0.0633	0.0027	−0.021	−0.0177	0.0954	−0.1053	0.0017	0.0238	−0.0015	−0.0004	−0.0142		−0.0028
PM2.5	−0.0271	−0.1379	0.0037	0.1108	0.0317	0.1987	−0.1308	0.0149	−0.1071	0.0062	0	0.0959	0.0012

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
