# Peer review of "In the Subtropical Monsoon Climate High-Density City, What Features of the Neighborhood Environment Matter Most for Public Health?"

_ijerph, 2020, doi:10.3390/ijerph17249566_

Round 1
Reviewer 1 Report
This study concerns very important issue related to urban areas: the impact of environmental conditions on the health of urban residents. The authors rightly point to the increasingly lethal impact of non-communicable diseases, many of which originate from environmental factors. Therefore, the stated goal of the paper in studying the link between public health and environmental factors in urban areas, and specifically in Guangzhou, China, is quite significant. However, there are some problems with the study, which I outline below.
1. Main research goal and literature review.
I am a bit confused about the specific purpose of the study. While the authors point to environmental factors and their impact on public health, they also use the term “built environment,” which relates to the manmade urban surroundings. Air quality is not part of the urban “built environment;” temperature, humidity, wind, etc. are not part of the “built environment” but rather they are all part of the natural environment. The built environment are the buildings, parks, parking spaces, anything that is constructed by human effort.
The authors need to make sure to use this terminology correctly.
In addition, there is no literature review section in the paper. There needs to be one that considers previous work related to the many variables used in the study. Have there been previous studies and what are their findings? There are 9 health dimensions proposed in the study and we don’t know anything about how they have been used in prior scholarly work. The paper needs to have a literature review section, where relevant studies about all measures used in the paper need to be incorporated and analyzed.
2. Methodology.
The authors do a good job explaining the selection of Guangzhou, which is one of the major Chinese cities. They also describe how they have conducted the measurements of all the public health related variables and neighborhood conditions. I think that it is valuable to have a study based on original data that was specifically collected by the authors. I also appreciate the specific attention to reliability and validity measures.
However, the explanation of the statistical methodology applied in the paper is seriously deficient. Neither the Analysis of Variance nor the Path modeling strategies are properly explained. For both of these strategies there are a number of assumptions that need to be studied and satisfied before we can conclude that the results are sound. There needs to be an explanation about why the variables the authors include in the model have been selected and what other variables might be missing. Is there specification bias? Heterogeneity? There are a number of conditions for each of these models that need to be met.
For example, several assumptions need to be met in order for a mixed model ANOVA to generate unbiased estimates of the main and interaction effects. One assumption is that the
residuals in both the between-subjects model and the within-subjects model
must be normally distributed.
The second assumption is homogeneity of variances or homoscedasticity. If the authors have performed some transformations of the data that may correct violations of these assumptions, but this needs to be discussed in the paper.
Another assumption is homogeneity of the variance-covariance matrices. The assumption is tested by Box’s M statistic. If Box’s M returns a p value that is less than .001, then the variance-covariance assumption is violated. Violations of this assumption can be corrected for with data transformations.
Yet another assumption of mixed model ANOVAs is “sphericity” and applies to
models including within-subjects variables with three or more levels. Mauchly’s test of sphericity can be used to evaluate this assumption.
Similarly, there are specific assumptions to be met regarding path analysis, variables cannot just be thrown together without careful consideration of such assumptions.
I would recommend that the authors carefully revise the paper with these comments in mind.
Author Response
Response to Reviewer 1 Comments
Comment 1: Main research goal and literature review.
I am a bit confused about the specific purpose of the study. While the authors point to environmental factors and their impact on public health, they also use the term “built environment,” which relates to the manmade urban surroundings. Air quality is not part of the urban “built environment;” temperature, humidity, wind, etc. are not part of the “built environment” but rather they are all part of the natural environment. The built environment are the buildings, parks, parking spaces, anything that is constructed by human effort.
The authors need to make sure to use this terminology correctly.
In addition, there is no literature review section in the paper. There needs to be one that considers previous work related to the many variables used in the study. Have there been previous studies and what are their findings? There are 9 health dimensions proposed in the study and we don’t know anything about how they have been used in prior scholarly work. The paper needs to have a literature review section, where relevant studies about all measures used in the paper need to be incorporated and analyzed.
Response1: Thanks for the reviewer’s suggestion. The “built environment” that appeared in most places was replaced with the “neighborhood environment” in the text. And a part of literature review on neighborhood environmental factors was added in the second paragraph of the introduction.
Comment 2. Methodology.
The authors do a good job explaining the selection of Guangzhou, which is one of the major Chinese cities. They also describe how they have conducted the measurements of all the public health related variables and neighborhood conditions. I think that it is valuable to have a study based on original data that was specifically collected by the authors. I also appreciate the specific attention to reliability and validity measures.
However, the explanation of the statistical methodology applied in the paper is seriously deficient. Neither the Analysis of Variance nor the Path modeling strategies are properly explained. For both of these strategies there are a number of assumptions that need to be studied and satisfied before we can conclude that the results are sound. There needs to be an explanation about why the variables the authors include in the model have been selected and what other variables might be missing. Is there specification bias? Heterogeneity? There are a number of conditions for each of these models that need to be met.
For example, several assumptions need to be met in order for a mixed model ANOVA to generate unbiased estimates of the main and interaction effects. One assumption is that the residuals in both the between-subjects model and the within-subjects model must be normally distributed.
The second assumption is homogeneity of variances or homoscedasticity. If the authors have performed some transformations of the data that may correct violations of these assumptions, but this needs to be discussed in the paper.
Another assumption is homogeneity of the variance-covariance matrices. The assumption is tested by Box’s M statistic. If Box’s M returns a p value that is less than .001, then the variance-covariance assumption is violated. Violations of this assumption can be corrected for with data transformations.
Yet another assumption of mixed model ANOVAs is “sphericity” and applies to models including within-subjects variables with three or more levels. Mauchly’s test of sphericity can be used to evaluate this assumption.
Similarly, there are specific assumptions to be met regarding path analysis, variables cannot just be thrown together without careful consideration of such assumptions.
Response2: Thanks for the reviewer’s suggestion. In the early stage of data analysis, we did a normality test on the dependent variables and residuals, to meet the assumptions for a mixed model ANOVA. The results are shown in Figure 1 and Figure 2. They basically present a normal distribution, and the outlierTest confirmed this (No Studentized residuals with Bonferroni p < 0.05).
|
|
|
Figure 1. Normality test on the dependent variables
|
|
|
|
Figure 2. Normality test on the residuals
Based on the reviewers’ opinions, we tested the homogeneity and “sphericity”. The result is shown in table 1, table 2 and table 3. According to the results, we can indicate that all of the variances and covariance between neighborhoods are consistent with the assumption of homogeneity (P>0.001), and assumption of “sphericity” is satisfied (P>0.1).
Table 1. Bartlett test of homogeneity of variances
|
Variables |
Group |
Bartlett's K-squared |
p-value |
|
SFTOTAL |
Neighborhood |
5.1651 |
0.1601 |
|
Season |
19.761 |
0.0055 |
|
|
HI1 |
Neighborhood |
3.7712 |
0.2872 |
|
Season |
23.415 |
0.0011 |
|
|
HI2 |
Neighborhood |
2.2183 |
0.5284 |
|
Season |
4.8766 |
0.3002 |
Table 2. Box's M-test for Homogeneity of Covariance Matrices
|
Group |
Chi-Sq (approx.) |
p-value |
|
Neighborhood |
40.671 |
0.00169 |
|
Season |
104.97 |
4.219e-12 |
Table 3. Mauchly's test of sphericity(Contrasts orthogonal to Season, Contrasts spanned by: Neighborhood + Season)
|
Variables |
W |
p-value |
|
SFTOTAL |
0.94801 |
0.3173 |
|
HI1 |
0.97006 |
0.5202 |
|
HI2 |
0.91219 |
0.1386 |
We thought that the sample size was large enough, so we did not include these results in the manuscript. As for the path analysis, we thought that as long as the data meets the above requirements, it should meet the assumptions of path analysis.

Reviewer 2 Report
The work entitled “In the hot and humid high-density city, what features of the built environment matter most for public health?” it presents a relationship between the public health of the neighborhood and the external environment built in old and high-density urban neighborhoods in a hot and humid area. The research is relevant and presents a topic of great importance, originality and scientific significance of the content, however, it needs to be improved in some aspects.
- in the abstract: the abstract is unclear and does not present information that invites the reader to explore the work. Define in the abstract which are the 10 environmental factors measured, as well as what is the SF-36 health scale and which are the 9 dimensions studied.
- the introduction explores little references and works of great importance to the theme addressed. An expanded expansion of the introduction text is needed, as well as bringing more references to solidify the theoretical basis of the research.
- as for materials and methods, there is only one observation to be made: in relation to the figures, all are of low resolution and do not present the geographic coordinates of the locations. It is necessary to correct.
- in relation to the results, in the discussion item, as well as in the introduction, there are few references to solidify the discussion on a theoretical basis. it is necessary to have a more comprehensive discussion of the results obtained with the results of works already published, which are rarely addressed in the work. It is necessary to compare the findings with the findings of other authors for other regions of similar climate, as well as for the same region, if researchs has already been carried out.
- in relation to the conclusions: it is necessary that the authors point out the restrictions and the strengths and weaknesses of the research. It is also necessary to indicate the possibilities for future research that can be carried out, especially in relation to what was not explored in this work.
After these corrections I recommend the publication.
Author Response
Response to Reviewer 2 Comments
Comment 1: In the abstract: the abstract is unclear and does not present information that invites the reader to explore the work. Define in the abstract which are the 10 environmental factors measured, as well as what is the SF-36 health scale and which are the 9 dimensions studied.
Response1: Thanks for the reviewer’s suggestion. In the abstract, in order to attract readers, we added a sentence: Taking Guangzhou as an example to explore the association of neighborhood environment and public health and preferably to offer some implications for better future city development. The 10 environmental factors and 9 health dimensions have been defined in the abstract.
Comment 2: the introduction explores little references and works of great importance to the theme addressed. An expanded expansion of the introduction text is needed, as well as bringing more references to solidify the theoretical basis of the research.
Response2: A part of literature review on neighborhood environmental factors was added in the second paragraph of the introduction.
Comment 3: as for materials and methods, there is only one observation to be made: in relation to the figures, all are of low resolution and do not present the geographic coordinates of the locations. It is necessary to correct.
Response3: Figure 1 and Figure 2 was revised and replaced with high-definition version, while the latitude and longitude lines, as well as geographic coordinates had been added.
Comment 4: in relation to the results, in the discussion item, as well as in the introduction, there are few references to solidify the discussion on a theoretical basis. it is necessary to have a more comprehensive discussion of the results obtained with the results of works already published, which are rarely addressed in the work. It is necessary to compare the findings with the findings of other authors for other regions of similar climate, as well as for the same region, if researchs has already been carried out.
Response4: The discussion 4.1 had been expanded. In 4.1 and 4.2, the comparison with the research results of the same area had been added, and the literature support of other research results had also been added. The order of cited literatures had also been adjusted.
Comment 5: in relation to the conclusions: it is necessary that the authors point out the restrictions and the strengths and weaknesses of the research. It is also necessary to indicate the possibilities for future research that can be carried out, especially in relation to what was not explored in this work.
Response5: Thanks for the reviewer’s suggestion. We have point out the restrictions and the strengths and weaknesses of the research, as well as the possibilities for future research.
Reviewer 3 Report
The article has scientific merit and a theme of extreme relevance. There’s no problem with the structure of the article and the data treatment is robust enough, so I congratulate the authors. However, I make observations that should be taken into account to improve its presentation.
In the title, the use of the expression "Hot and humid" makes no sense. Consider the climate classification you have used and refer to subtropical monsoon climate.
Lines 67 and 68 - the factors considered are based on which references and authors? Mention them, please.
Line 74 and on location maps – it’s insufficient to mention the location if the grid lines of the maps in figures 1 and 2 informing latitude and longitude are missing. Please insert this information on the maps. It is a matter of cartographic convention.
I observed that frequently the figures, graphs and tables appear before their reference in the text making it very difficult to read. This happened with figures 2, 3 and 4. I noticed that figure 3 is not contextualized in the text. Also, in my opinion, figure 4 is actually a graph. Please correct these points.
Line 271 - Urban heat island is a concept and needs reference.
Line 312 - How do you conclude the relationship with COVID-19? This became uncertain in the text. Provide references to state this because you only referred to influenza [16].
Author Response
Response to Reviewer 3 Comments
Comment 1: In the title, the use of the expression "Hot and humid" makes no sense. Consider the climate classification you have used and refer to subtropical monsoon climate.
Response1: Thanks for the reviewer’s suggestion. We have changed the title into “In the subtropical monsoon climate high-density city,what features of the neighborhood environment matter most for public health?” in the revised manuscript (line 2 -4).
Comment 2: Lines 67 and 68 - the factors considered are based on which references and authors? Mention them, please.
Response2: A part of literature review on neighborhood environmental factors was added in the second paragraph of the introduction.
Comment 3: Line 74 and on location maps – it’s insufficient to mention the location if the grid lines of the maps in figures 1 and 2 informing latitude and longitude are missing. Please insert this information on the maps. It is a matter of cartographic convention.
Response3: Figure 1 and Figure 2 was revised and replaced with high-definition version, while the latitude and longitude lines, as well as geographic coordinates had been added.
Comment 4: I observed that frequently the figures, graphs and tables appear before their reference in the text making it very difficult to read. This happened with figures 2, 3 and 4. I noticed that figure 3 is not contextualized in the text. Also, in my opinion, figure 4 is actually a graph. Please correct these points.
Response4: Figure 3, which had little relation to the article, had been deleted. Figure 4 had been changed to a table (Table 2). The subsequent icon numbers had also been adjusted.
Comment 5: Line 271 - Urban heat island is a concept and needs reference.
Response5: Thanks for the reviewer’s suggestion. The reference has been added.
Comment 6: Line 312 - How do you conclude the relationship with COVID-19? This became uncertain in the text. Provide references to state this because you only referred to influenza [16].
Response6: the “COVID-19” has been deleted, and reference about influenza has been added.
Round 2
Reviewer 1 Report
1. Main research goal and literature review.
I appreciate the author’s attempt to amend the literature review and that they decided to change “built environment” to “neighborhood environment.” That change alone signifies the complete lack of theoretical conceptualization in the paper. What is a “neighborhood environment”? What constitutes it? What are the parameters of it? What is the definition? How does it relate to the built environment or any other urban concept?
The authors demonstrate a complete lack of understanding how to situate their data within the urban and environmental studies literature. There are no proper research questions identified in the paper, there are no hypotheses, there are no competing explanations about anything.
I recommend that the author reads some of the papers, already published in the journal, for example, the most recent paper: “Is Family Structure Associated with Deviance Propensity during Adolescence? The Role of Family Climate and Anger Dysregulation.”
2. Methodology.
I appreciate the histograms that the authors put in the reply as evidence of the normal distribution of their variables. In my opinion, none of these variables are normally distributed. All of them are left-skewed.
There is still no explanation of the selection of mixed method ANOVA, there is no explanation of selecting path analysis, let alone a conceptual model for the Analysis of Variance or for the path analysis. Path analysis is among the most complex statistical modeling techniques and there is not a word of explanation about it. I would strongly recommend Morgan and Winship’s book (2015): Counterfactuals and Causal Inference: Methods and Principals for Social Research. Second Edition. Cambridge University Press.